

# Heavy snow loads in Finnish forests respond regionally asymmetrically to projected climate change

Ilari Lehtonen[1], Matti Kämäräinen[1], Hilppa Gregow[1], Ari Venäläinen[1], Heli Peltola[2]

[1]Finnish Meteorological Institute, Helsinki, FI-00101, Finland
[2]School of Forest Sciences, University of Easteren Finland, Joensuu, FI-80101, Finland

*Correspondence to*: Ilari Lehtonen (ilari.lehtonen@fmi.fi)

**Abstract.** This study examined the impacts of projected climate change on heavy snow loads on Finnish forests, where snow-induced forest damage occurs frequently. We used for snow-load calculations daily data from five global climate models under representative concentration pathway (RCP) scenarios RCP4.5 and RCP8.5, as statistically downscaled onto a high-resolution grid using a quantile-mapping method. Our results suggest that projected climate warming results in regionally asymmetric response on heavy snow loads in Finnish forests. In eastern and northern Finland, the annual maximum snow loads on tree crowns were projected to increase during the present century, opposite to southern and western parts of the country. The change was rather similar both for heavy rime loads and wet snow loads, as well as for frozen snow loads. Only the heaviest dry snow loads were projected probably to decrease almost over the whole of Finland. Our results accord with previous snowfall projections indicating typically increasing heavy snowfalls over the areas with mean temperature below −8 °C. In spite of some uncertainties related to our results, we conclude that the risk for snow-induced forest damage is likely to increase in the future in the eastern and northern parts of Finland, i.e. in the areas experiencing the coldest winters in the country. The increase is partly due to increase in wet snow hazards but also due to more favourable conditions for rime accumulation in more humid but still cold enough future climate.

## 1 Introduction

Forest damage caused by snow loading on trees occurs frequently in boreal environments. On the European level, estimates about the amount of timber damaged by snow during a typical year vary between one and four million cubic meters (Nykänen et al., 1997; Schelhaas et al., 2003). In Finland, insurance companies have paid within the last three decades on average approximately 0.5 M€ compensation annually due to snow damage of forests, which accounts about 7% of the total indemnities paid for forest owners (Finnish Forest Research Institute, 2014). Thus, snow is among the most important abiotic stress factors in the Finnish forests after windstorms which account about 77% of the forest damage compensated by private insurance companies. In addition, snow damaged trees occasionally disrupt seriously power transmission by bending or leaning over power lines. Trees damaged by snow are furthermore susceptible to insect attacks and other kind of consequential damage (e.g., Schroeder and Eidmann, 1993; Schlyter et al., 2006).



The risk of snow damage is strongly dependent upon weather and climatological factors. Temperature influences the moisture content and snow attaches on tree crowns and branches most effectively when temperature at the time of precipitation is close to the freezing point (Solantie, 1994). Moderate wind speeds enhance the snow accumulation but strong winds dislodge most of the snow from the tree crowns. Topography also plays an important role in the occurrence of snow damage. The heaviest

snow loads tend to accumulate on forests at high altitudes (e.g., Jalkanen and Konôpka, 1998), largely because rime accumulation is most efficient on places located above the surrounding terrain (Makkonen and Ahti, 1995), but also because of orographic addition to precipitation.

Climate in northern Europe is projected to change considerably during the 21st century due to increasing greenhouse-gas concentrations in the atmosphere (e.g., Räisänen and Ylhäisi, 2015). In winter, both temperature and precipitation are projected

to increase with high confidence. This will evidently lead to changes in snow climate having multiple effects on ecological systems (Callaghan et al., 2011). In the coldest areas, snowfall is generally projected to increase (Krasting et al., 2013; Räisänen, 2015) while over milder regions, a considerably larger fraction of total precipitation is expected to fall as liquid form in a warmer climate. How this anticipated change will affect the risk of snow damage for trees is not straightforward given the sensitive nature of snow accumulation to specific weather conditions. Because different forest management options alter the

probabilities of various types of forest damage differently (Nykänen et al., 1997; Klopcic et al., 2009), accurate estimates for changes in conditions causing the damage would be beneficial for adaptation purposes (Jönsson et al., 2015).

Kilpeläinen et al. (2010) estimated that the risk for snow-induced forest damage would decrease throughout Finland towards the end of the present century whereas Gregow et al. (2011) argued that because of decreasing soil frost supporting tree anchorage and increasing heavy snow loads, the risk for uprooting would increase in southern Finland and the risk for stem

breakage in northern Finland In both studies, a cumulative snow-load model presented by Gregow et al. (2008), referred to hereinafter as G08 method, was applied. The G08 method estimates the amount of snow load on tree crowns by applying precipitation, temperature and wind speed observations as input. However, Lehtonen et al. (2014) showed that when the snow load is classified into different components (rime, dry snow, wet snow and frozen snow), the snow loads simulated by the G08 method correlate best with dry snow loads having little importance with regard to the forest damage. Therefore, it would be

interesting to apply a more sophisticated method in studying the climate change impacts on heavy snow loads on tree crowns. This may also give a more in-depth look on the relative importance of different snow load types in the change.

In this study, we apply the snow load calculation and classification method used by Lehtonen et al. (2014), hereafter referred to FMI method, to assess the climate change impacts on the occurrence of heavy snow loads on tree crowns in Finland. We use in snow-load calculations daily data from five independent general circulation models (GCMs) participating in the Coupled

Model Intercomparison Project (CMIP) phase 5 (Taylor et al., 2012) under representative concentration pathway (RCP) scenarios RCP4.5 and RCP8.5 (van Vuuren et al., 2011) over the period 1980–2099. Before the snow-load calculations, the modelled values of used weather variables were downscaled onto a high-resolution $0.1° \times 0.2°$ latitude–longitude grid covering Finland by using a quantile-mapping technique. The comparison of results based on different GCMs instead of multi-model



mean approach enabled assessing the model-based uncertainty in the climate change response because different models simulate different changes in climate in response to the same radiative forcing.

## 2 Materials and methods

### 2.1 Climate data

We used daily data from five CMIP5 models listed in Table 1 over the period 1980–2099. The models were chosen on the basis of their skill to simulate present-day average monthly temperature and precipitation climatology in northern Europe and the availability of all required variables on a daily time scale. The variables used in this study were mean, maximum and minimum air temperature at 2 m height, mean relative humidity at 2 m height, mean wind speed at 10 m height and total precipitation. Historical simulations until 2005 were combined with simulations under RCP4.5 and RCP8.5 emission scenarios

for the period 2006–2099. The RCP8.5 is a high-emission scenario leading to about twice as fast global warming than the more modest RCP4.5 scenario (Collins et al., 2013).

Before further data analysis, we performed a combined statistical downscaling and bias correction to the model data by applying a quantile mapping technique using smoothing. The model data were downscaled onto a regular $0.1° \times 0.2°$ grid (approximately 10 km $\times$ 10 km) covering Finland. In quantile mapping, cumulative probability distributions of simulated time

series of weather variables are fitted to the observed distributions within the calibration period (1981–2010 in our case), separately for each month. For our observational data set, we used gridded data interpolated from the station observations made by the Finnish Meteorological Institute (FMI). The interpolation of station observations was done by applying kriging with external drift (Aalto et al., 2013). However, for wind speed, the quality of observations did not support the creation of homogenous, gridded daily data set over Finland; therefore we used daily wind speeds from the European Center for Medium-

Range Weather Forecast ERA-Interim reanalysis (Dee et al., 2011) provided on a regular $0.75° \times 0.75°$ grid. The wind speeds were bilinearly interpolated onto the same $0.1° \times 0.2°$ grid with other variables but the true resolution was still much coarser for wind speed than for the other variables.

A detailed evaluation of quantile mapping for correcting simulated temperature time series was presented by Räisänen and Räty (2013) and for correcting simulated precipitation time series by Räty et al. (2014). Wilcke et al. (2013) furthermore

demonstrated the use of quantile mapping for correcting also relative humidity and wind speed simulations of regional climate models. The effect of bias correction on simulated distributions of weather variables is exemplified in Fig. 1. For temperature, precipitation and wind speed the corrected distributions correspond to the observed distributions within the calibration period by definition. Within scenario periods, the same corrections are applied to the simulated values than within the calibration period. Note that the peak in temperature distribution near the freezing point is retained both during the calibration and scenario

periods whereas the use of a delta-change method would unrealistically shift the peak.

For relative humidity, the correction was less accurate in freezing temperatures. That is because relative humidity in subzero temperatures can be expressed either relative to ice or supercooled water. The only difference between these two formulations





is that the maximum possible water vapour content is assumed to be larger when humidity is expressed relative to supercooled water (Hardy, 1998). Hence, humidity of 100% relative to water in freezing temperatures would be over 100% relative to ice, but in nature these oversaturated situations with respect to ice rarely occur. For this reason, we performed the bias correction for humidities relative to ice to avoid nonphysical oversaturated situations. Because in station observations humidity is

expressed relative to water and in the model results relative to ice, we first transformed the observed humidities relative to ice following Hardy (1998). After the bias correction, the corrected distributions of humidity corresponded to the observed distributions relative to ice by definition. In the snow-load calculations, however, we needed the humidity to be expressed relative to water and we thus transformed the corrected humidities back to relative to water. After this final step, the corrected distributions of relative humidity did not necessarily correspond to the observed distributions. For some models the corrected

and observed distributions were still close to each other but for part of the models the deviation was larger. It appeared that the annual maximum rime loads calculated from the corrected model data were, on average, approximately 20% larger than those calculated from the observational data during the calibration period 1981–2010 (not shown). This indicates that after the corrections, there were typically too many humid days in freezing temperatures.

We also note that conventional relative humidity measurements in freezing temperatures are not typically desirably accurate

(Makkonen and Laakso, 2005). This was clear in our data as the observational relative humidities commonly had supersaturated values with respect to ice. However, after our bias correction, the humidity was not allowed to exceed 100% relative to ice. Moreover, part of the models had frequently unrealistic supersaturated humidities in subzero temperatures. We recognize that these inaccuracies in the observational humidity data and nonphysicalities in the model data were a source of uncertainty in this study.

Projected wintertime (November–March) changes in climate variables in our data set after the bias correction are displayed in Fig. 2. Projections are shown separately for southern, central and northern Finland (Fig. 3). The division of Finland into subregions follows that by Lehtonen et al. (2014). Temperature and relative humidity with respect to water are generally projected to increase towards end of the century, the increase being larger in the north than in the south. For the relative humidity this is partly because in subzero temperatures constant humidity relative to ice leads to increasing humidity relative

to water with increasing temperature. Nevertheless, in southern Finland, relative humidity may even decrease. For temperature, the projected increase until the end of the century is about 3–5 °C under the RCP4.5 and about 5–9 °C under the RCP8.5 scenario compared to the baseline period 1980–2099, depending on the model and region. Precipitation is projected to increase rather uniformly throughout the country but the uncertainty ranges in the rate of the change are fairly large. Nonetheless, the projected changes are larger under the RCP8.5 than RCP4.5 scenario. Based on multi-model mean, mean wind speed is

projected to stay almost unaltered. However, considerable variability in the projections exists between the models and one model suggests that mean wind speed could increase even by over 30% in central Finland.



## 2.2 Crown snow load calculations

We modelled the snow load amounts on tree crowns by applying the FMI method. This method has been developed at FMI for operational purposes as an early-warning tool for heavy snow loads on tree crowns threatening mainly the functionality of electric power network. A detailed description of the method was presented by Lehtonen et al. (2014) with discussion of the

climatology of heavy crown snow loads in Finland. In a nutshell, the modelled snow load is classified into four different types: rime, dry snow, wet snow and frozen snow. Decrease of the modelled snow load may occur due to wind removal or melting and the snow may also change its type. For instance, wet snow may be transformed into frozen snow. Increase of the snow load is caused by accumulation of rime and snowfall. Rime is the only snow load type in the model affected by relative humidity. In producing the crown snow load climatology, Lehtonen et al. (2014) excluded the effects of solar radiation and

cloudiness in the model because lack of sufficient observations. Here, we followed the same approach, which leads in some cases to too intense riming and underestimation of snow removal in late winter (Lehtonen et al., 2014).

The FMI method has a time step of one hour but according to Lehtonen et al. (2014), the snow load amounts calculated with daily mean data correlate on average rather well with those calculated based on hourly or 3-hourly data, particularly in the case of rime and dry snow. In this study, we used only daily data but we mimicked diurnal cycle of 2-m temperature by assuming

in the calculations daily minimum temperature to occur at 00 UTC and maximum temperature at 12 UTC. Similarly, we assumed that daily mean temperature prevailed at 06 and 18 UTC and that temperature changed linearly between these four moments. Other weather variables were assumed to stay constant throughout a day. Although the dependence between temperature and time of day in Finland in winter is weak, the artificially induced temperature cycle was expected to improve the results when temperature varied around the freezing point. Kilpeläinen et al. (2010) used a similar approach in their work

by utilizing the sine function in daily temperatures.

In addition to weather variables, the riming efficiency in the FMI model is influenced by topography. The modelled effectiveness of riming reaches its maximum value at elevation of 400 meters above the mean sea level. The areas affected by enhanced riming include many areas in northern Finland and northeastern central Finland (Fig. 3).

In order to compare our results to those by Kilpeläinen et al. (2010), we calculated the snow loads also with the same G08

method (Gregow et al., 2008) that was used by Kilpeläinen et al. (2010). Compared to the FMI method, the G08 method is computationally far simpler. The main deficiency of the G08 method is the exclusion of riming as especially at high altitudes in northern Finland rime accretion is the most important factor leading to heavy crown snow loads.

We also calculated the number of risk days for heavy rime and total snow load accretion based on daily mean values of 2-m air temperature, 2-m relative humidity, 10-m wind speed and precipitation. Here we used the thresholds defined by Lehtonen

et al. (2014), although we recognize that these thresholds may not be well suited for the whole country. Nonetheless, according to Lehtonen et al. (2014), the geographical distribution of number of risk days for heavy snow loading by using these thresholds corresponds well with the geographical distribution of modelled heavy wet snow loads. For the risk days for heavy riming, the



same kind of dependence did not hold true. Moreover, the number of days expressing heavy rime accumulation was underestimated on coastal regions and at high altitudes based on the number of risk days.

## 3 Results

The annual maximum crown snow loads in Finland as averaged over the period 1981–2010 are shown in Fig. 4. The results are calculated from the observational daily data using both the FMI and G08 methods and the values are shown for the total crown snow load as well as for the different snow load types of the FMI method. The numbers of risk days for heavy snow loading and riming are displayed as well. The modelled annual maximum snow loads tend to be generally slightly heavier based on the FMI method than G08 method, especially over the high-elevated areas where the modelled rime loads are heaviest.

The same variables for the future period 2070–2099 under the high-emission RCP8.5 scenario are shown in Fig. 5 as a multi-model mean, along with percentage changes to the period 1980–2009. The annual maximum rime loads, as well as wet snow and frozen snow loads are projected to increase in eastern and northern Finland up to 60% compared to the period 1980–2009. In southern and western Finland, the annual maximum snow loads are simultaneously projected to decrease. Projected changes for total snow load based on the FMI method resemble closely those for rime, wet snow and frozen snow loads. For dry snow,

the annual maximum loads are projected to decrease almost in whole of Finland. Only in northern Finland they are projected to remain roughly unaltered. The situation is similar for total snow loads calculated by using the G08 method. Risk days for heavy snow loading are projected to change rather similarly than heavy wet snow loads. For intense riming, the favourable conditions are projected to become much more common in eastern and northern Finland.

The projected changes in the annual maximum values of different snow load components for all the three future periods 2010–

2039, 2040–2069 and 2070–2099 compared to the period 1980–2009 are displayed in Fig. 6 as areal averages over southern, central and northern Finland. For the near-future period 2010–2039, the projected changes are small except in northern Finland, where annual maximum rime, wet snow and frozen snow loads are already projected to increase by about 10–30%. For the more distant periods, not much additional increases in northern Finland are expected but in the south, the annual maximum crown snow loads are projected to start to decrease. In central Finland, the projected changes are on average rather small for

all time periods and snow load components. The eastern parts of the region are nevertheless experiencing mainly increasing and western parts decreasing trends (Fig. 5). However, the zero line in the projected changes compared to the period 1980–2009 is in general slowly moving towards northeast when moving to more distant future periods (not shown), Fig. 5 representing the most distant future period considered in this study. This nonlinearity of the changes can be seen in Fig. 6. For instance, under the RCP8.5 scenario, the heavy wet snow loads and frozen snow loads are likely to slightly increase even in

southern Finland at first but towards the end of the 21st century, they are projected to decrease by about 10–40%. Similarly, the dry snow loads are at first projected to slightly increase in the north but rather decrease thereafter. Fig. 6 moreover confirms



the conclusion that projected changes in the total crown snow loads estimated by the G08 method resemble mostly those of dry snow loads by the FMI method.

The uncertainty in the projections related to the model choice is fairly modest (Fig. 6). All models suggest the annual maximum snow loads to increase in the north with the exception of dry snow loads and total snow loads by the G08 method. Similarly, the annual maximum loads of all snow load types are projected to decrease in the south by all models by the end of the present century.

The numbers of risk days for heavy rime and total snow loading are mostly projected to decrease both in early and late winter (Fig. 7). During midwinter months, the numbers of risk days are projected to remain nearly unaltered in southern Finland, to increase slightly in central Finland and to increase considerably in northern Finland. Early winter would still be the most favourable time of year for heavy rime and snow accretion. But, for example, in northern Finland, December might be the most favourable month for riming in the future instead of November and wet snow hazards could occur most frequently in November and December instead of October and November.

## 4 Discussion

There are many sources of uncertainty in our results. Firstly, the modelled amounts of snow loads on tree crowns are highly sensitive to the weather conditions near the freezing point. Hence, it is clear that the use of uncorrected model data being somewhat biased would not have been desirable in this study. Similarly, the feasibility of the applied bias correction method to our purposes is utmost important. We applied quantile mapping that is widely adopted technique in climate change impact studies and generally suggested (Teutschbein and Seibert, 2012). It has moreover proven to be among the best-performing empirical bias-correction methods for temperature (Räisänen and Räty, 2013) and precipitation (Räty et al., 2014) throughout the probability distribution. On the other hand, bias correction alters spatiotemporal relations between different variables without satisfactory physical justification (Ehret et al., 2012). This adds uncertainty to our results as they are sensitive to a suitable combination of several weather variables. Most uncertain the bias correction was for relative humidity. On the other hand, rime was the only snow type in the model affected by relative humidity.

Other noteworthy sources of uncertainty are the snow load models themselves. The deficiencies related to the applied methods were discussed earlier by Lehtonen et al. (2014). Again, reasonable modelling of rime loads was considered to be particularly uncertain. One reason for this was that properties of rime are sensitive to the formation conditions (e.g., Wang and Jiang, 2012). Additionally, in this study, the use of daily data was assumed to deteriorate the results. However, with the exception of considerably heavier rime loads in the north-westernmost Finland, the geographical distributions of modelled annual maximum snow loads resembled closely those presented by Lehtonen et al. (2014) by using 3-hourly station observations as input data. This suggests that climatological features of heavy crown snow loads were satisfactorily captured by using daily weather data. In fact, Lehtonen et al. (2014) hypothesized that the modelled rime loads were underestimated at high elevations in northern



Finland due to too efficient wind removal. While the resolution of our wind speed data was coarser compared to other variables, these deficiencies overruled each other in this study and led possibly in fact to more realistic simulation of rime loads in northern Finland. We further note that the heaviest rime loads were modelled over approximately the same areas which were recognized to be most susceptible for losses in wind energy production in Finland due to icing (Ljungberg and Niemelä, 2011).

To estimate the model-based uncertainty related to the projections, we used data from five different climate models. The different models proved to produce fairly similar response on the occurrence of heavy crown snow loads consisting mainly increase in the annual maximum snow loads in eastern and northern Finland and decrease in southern and western Finland. While the projections for rime loads are considered to be most uncertain, it is worth of noting that the projected changes for heavy wet and frozen snow loads had rather similar geographical characteristics as those for the rime loads. This reinforces

the understanding that the risk for snow-induced forest damages is likely to rather increase than decrease in the eastern and northern parts of the country. This contradicts to the results of Kilpeläinen et al. (2010) suggesting decreasing risk for snow damages throughout Finland towards the end of this century but is in better agreement with the conclusions of Gregow et al. (2011). The lack of division of snow load into different types and complete omitting of riming is expected to be the major deficiency in the G08 method used by Kilpeläinen et al. (2010) and Gregow et al. (2011). In addition, Kilpeläinen et al. (2010)

did not use model data for wind speed but assumed the wind speed to stay constant. Our results further confirmed the conclusion of Lehtonen et al. (2014) that the total snow load simulated by the G08 method correlates better with dry snow load than other snow load types of the FMI method. Moreover, the projected changes for both the annual maximum dry snow loads and the total snow loads by using the G08 method in this study were rather similar to those presented by Kilpeläinen et al. (2010). In overall, the differences in our results compared to those of Kilpeläinen et al. (2010) are attributable to the different

calculation methods of snow loads while the climate change impact itself was fairly similar in these two studies.
While the risk for snow damage differs between different tree species and between managed and unmanaged stands (Päätalo, 2000), possible climate-change driven changes in tree species composition, like proposed by Kellomäki et al. (2008), may further affect to the overall snow-damage risk of forests. Similarly, proper forest management aids to control or decrease the risk.

Based on an ensemble of regional climate model simulations, Räisänen (2016) showed recently that snowfall in northern Europe is typically projected to decrease in a warming climate everywhere where mean temperature exceeds −11 °C. However, snowfall extremes occur in similar conditions with near-zero temperatures regardless of climate warming and thus changes in the frequency of those conditions are relevant to heavy snow loads. Hence, less systematic changes have been simulated for the intensity of extreme daily snowfall than for total snowfall (O'Gorman, 2014; Räisänen, 2016). As seen in Fig. 1, near-zero

temperatures may occur in the future as frequently as currently in spite of general warming. According to previous model studies (de Vries et al., 2014; Räisänen, 2016), the maximum snowfall decreases over the areas where mean temperature exceeds −8 °C. This is fairly well in accordance with our projections for heavy wet snow loads as they are projected to increase roughly over the area where mean temperature of midwinter months from December to February is in current climate below −8 °C (Pirinen et al., 2012). In further accordance with the present results, Räisänen (2016) also noted similar change in snowfall





seasonality that is apparent in the number of risk days for heavy snow loading: in northern Fennoscandia snowfall is projected to increase between December and March and to decrease during early and late winter.

## 5 Conclusions

The impact of projected climate change on heavy snow loads on tree crowns in Finland was studied in this work by using the statistically downscaled and bias-corrected daily output of five CMIP5 models. Our results indicate that while climate becomes warmer, the annual maximum snow loads are likely to increase in eastern and northern Finland while in the southern and western parts of the country they are expected to decrease. This implies that there exists an increasing need to consider risks of snow damage in forest management in eastern and northern Finland. Our results contradict to those by Kilpeläinen et al.

(2010) suggesting that the risk for snow damage would decrease in the whole of Finland towards end of this century. This difference is attributable to differences in used snow load calculation methods. By dividing the snow load into different components, namely rime, dry snow, wet snow and frozen snow, it was demonstrated that only dry snow loads are projected to change similarly with the projections presented by Kilpeläinen et al. (2010). Our results moreover accord with those of Räisänen (2016) showing that heavy snowfalls in northern Europe are likely to increase over the areas experiencing the coldest

winters.

Our results are affected by many sources of uncertainty. The main challenge in modelling crown snow loads is that the accumulation of snow and rime is sensitive to small variations in multiple weather variables. For instance, snowfall accumulates on tree branches and crowns most effectively within a narrow temperature range around 0 °C. Hence, our results are fairly sensitive to the applied bias-correction procedures. The modelling of rime loads was considered to be most uncertain,

mostly due to uncertainties related to correcting of relative humidity. However, the projected changes for the annual maximum rime loads proved to be fairly similar than those for wet and frozen snow loads. Hence, we believe that these results will serve as a valuable basis for anticipated changes in risks for snow damage in Finnish forests.

## Acknowledgments

This research has been supported by the Consortium project ADAPT (Adaptation of forest management to climate change:

uncertainties, impacts, and risks to forests and forestry in Finland), which is a collaboration project between University of Eastern Finland (UEF proj. 14907) and FMI (proj. 260785) and funded jointly by the Academy of Finland, UEF and FMI. Support was also received from the Strategic Research Council at the Academy of Finland through the FORBIO (Sustainable, climate-neutral, and resource-efficient forest-based bioeconomy) project. We acknowledge the World Climate Research Programme's Working Group on Coupled Modelling, which is responsible for CMIP, and we thank the climate modeling

groups (listed in Table 1 of this paper) for producing and making available their model output. For CMIP the U.S. Department



of Energy's Program for Climate Model Diagnosis and Intercomparison provides coordinating support and led development of software infrastructure in partnership with the Global Organization for Earth System Science Portals. The ERA-Interim data were obtained from the European Center for Medium-Range Weather Forecasts data server. The development of the FMI method is based on the work of Petri Hoppula, Reijo Solantie, Mika Heiskanen, Kari Ahti, Bengt Tammelin and Kristiina

Säntti at FMI. We thank Kimmo Ruosteenoja for downloading and preprocessing the model data. Pentti Pirinen is acknowledged for interpolating the observational data onto the Finnish grid. We appreciate Olle Räty and Jouni Räisänen from Department of Physics, University of Helsinki, for developing the applied bias correction software.

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

**Table 1. CMIP5 models used in this study with information on country of origin and resolution of the models (L refers to number of vertical levels, T to triangular truncation and C to cubed sphere).**

| Model | Country of origin | Resolution (lon × lat), level | Reference |
|---|---|---|---|
| CanESM2 | Canada | T63 (1.875° × 1.875°), L35 | von Salzen et al. (2013) |
| CNRM-CM5 | France | T127 (1.4° × 1.4°), L31 | Voldoire et al. (2013) |
| GFDL-CM3 | United States | C48 (2.5° × 2.0°), L48 | Donner et al. (2011) |
| HadGEM2-ES | United Kingdom | 1.25° × 1.875°, L38 | Collins et al. (2011) |
| MIROC5 | Japan | T85 (1.4° × 1.4°), L40 | Watanabe et al. (2010) |


**Table 2. Threshold values of daily 2-m mean air temperature (Tmean), 2-m mean relative humidity (RHmean), 10-m mean wind speed (Umean) and total precipitation (Pday) determined by Lehtonen et al. (2014) for risk days favourable for heavy snow loading and heavy riming.**





| Risk days for heavy snow-loading | Risk days for heavy riming |
|---|---|
| $-3.42\ °C < T_{mean} < 1.05\ °C$ | $-5.19\ °C < T_{mean} < -0.16\ °C$ |
| $RH_{mean} > 89.44\%$ | $RH_{mean} > 95.50\%$ |
| $2.07\ \text{m s}^{-1} < U_{mean} < 5.63\ \text{m s}^{-1}$ | $2.00\ \text{m s}^{-1} < U_{mean} < 4.54\ \text{m s}^{-1}$ |
| $P_{day} > 6.41\ \text{mm}$ | $P_{day} < 1.11\ \text{mm}$ |

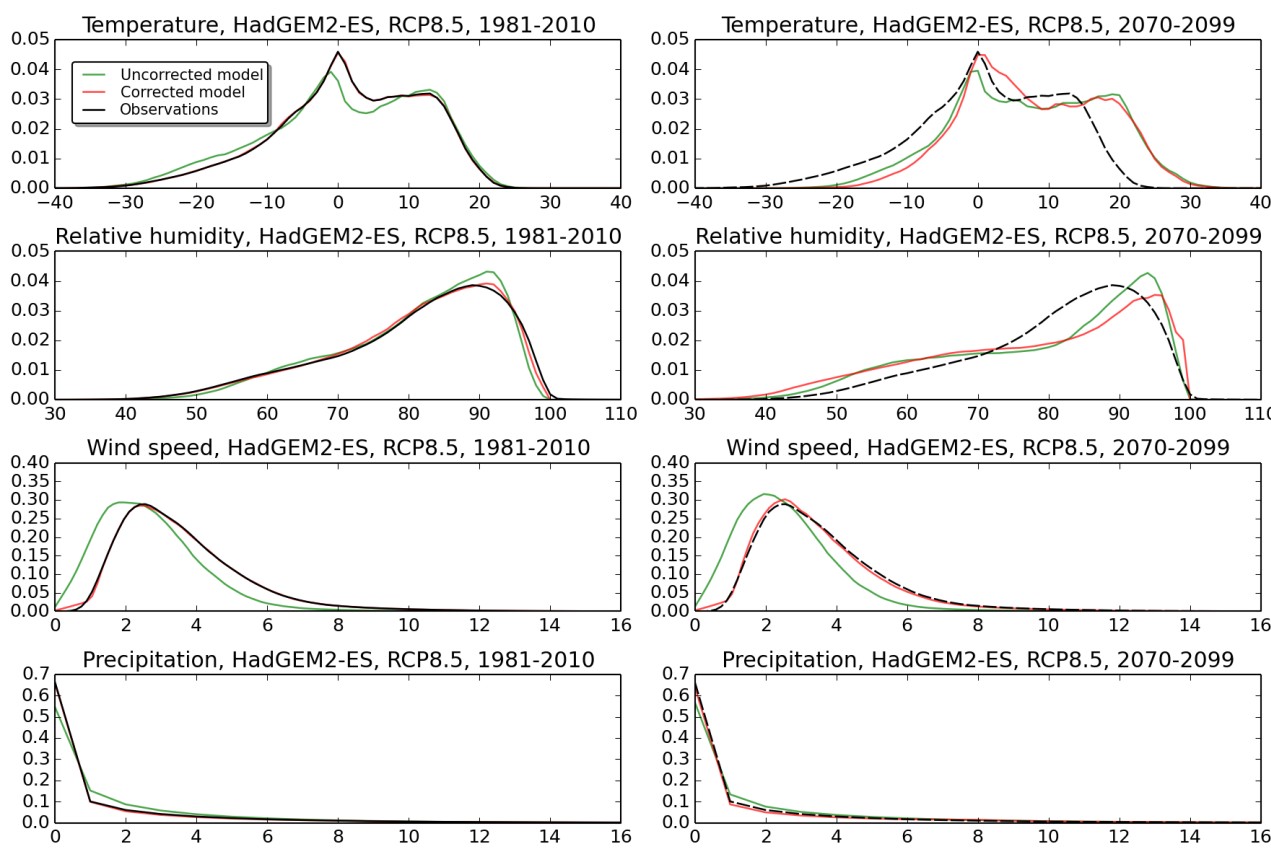

**Figure 1. Probability distributions of daily mean 2-m air temperature, daily mean 2-m relative humidity, daily mean 10-m wind speed and daily total precipitation in the HadGEM2-ES model for the periods 1981–2010 and 2070–2099 under the RCP8.5 scenario over the whole of Finland. Green curves represent the uncorrected model data and red curves the bias-corrected model data. Black curves represent the observational data during the calibration period 1981–2010. Dotted black curves display the observational probability distributions during the calibration period 1981–2010.**







**Figure 2. Projected changes in November–March mean 2-m air temperature (a), mean 2-m relative humidity (b), mean 10-m wind speed (c), and total precipitation (d) compared to the period 1980–2009 and averaged over southern, central and northern Finland. Dots indicate the multi-model mean change and whiskers extend to the maximum and minimum projections.**


**Figure 3. Division of Finland into southern, central and northern Finland. Shading shows the elevation above mean sea level.**



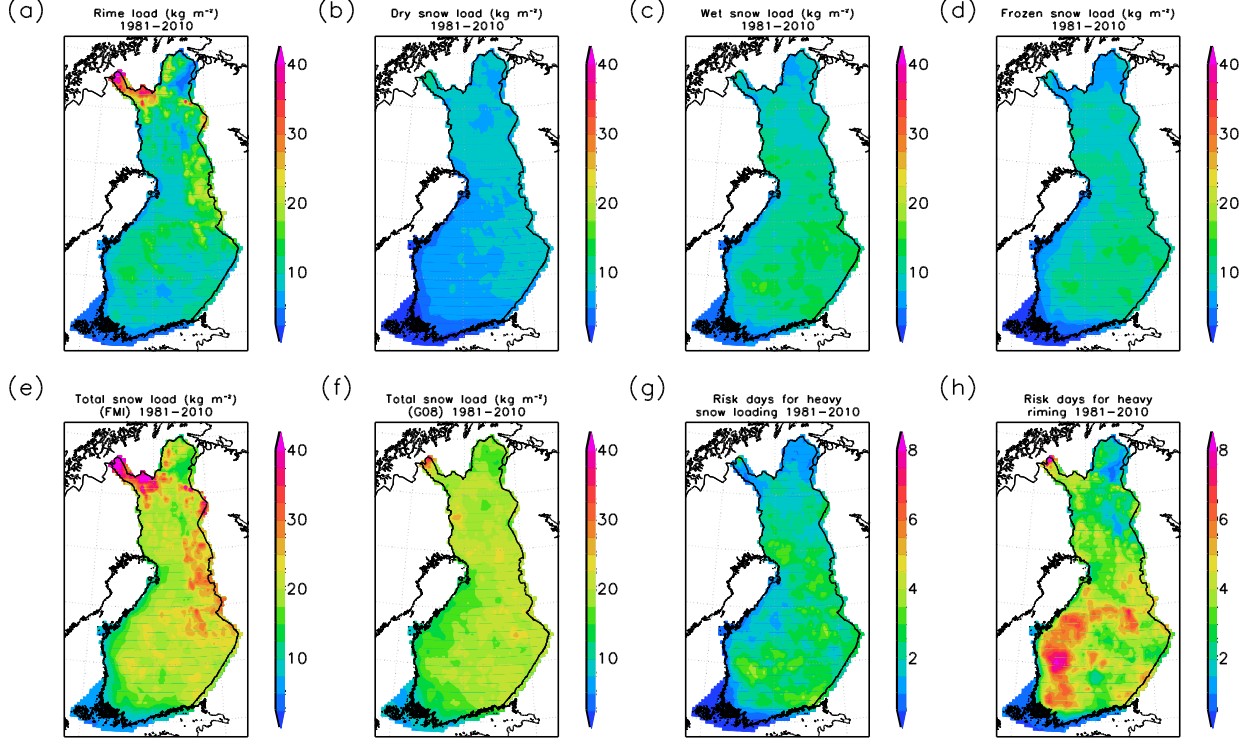

**Figure 4.** The annual maximum rime loads (a), dry snow loads (b), wet snow loads (c), frozen snow loads (d), total snow loads based on the FMI method (e), and total snow loads based on the G08 method (f) averaged over the period 1981–2010 and calculated from the observational weather data. The annual numbers of risk days for heavy snow loading (g) and heavy riming (h) are shown as well.





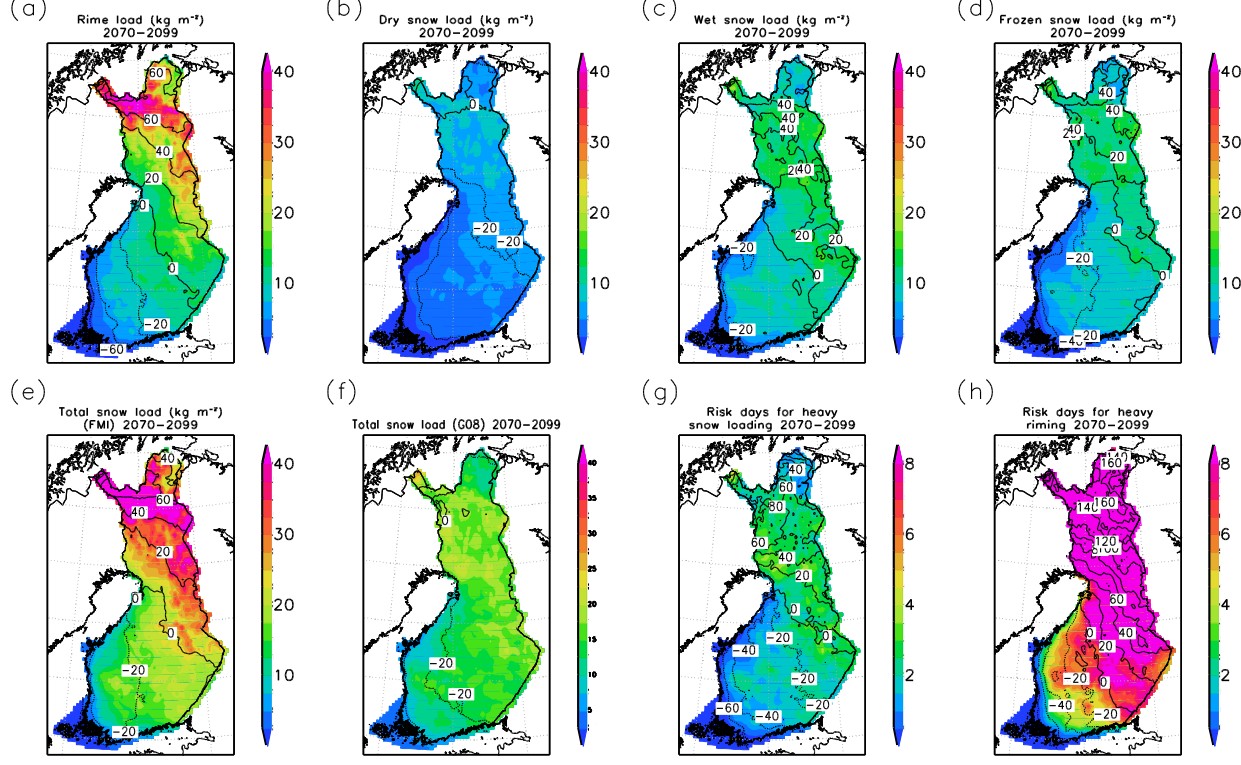

**Figure 5. The annual maximum rime loads (a), dry snow loads (b), wet snow loads (c), frozen snow loads (d), total snow loads based on the FMI method (e), and total snow loads based on the G08 method (f) for the period 2070–2099 under the RCP8.5 scenario as a multi-model mean. Contours show the multi-model mean change from 1980–2009 to 2070–2099.**





**Figure 6. Projected changes in the annual maximum rime loads (a), dry snow loads (b), wet snow loads (c), frozen snow loads (d), total snow loads based on the FMI method (e) and total snow loads based on the G08 method (f) compared to the period 1980–2009 and averaged over southern (S), central (C) and northern (N) Finland. Dots indicate the multi-model mean change and whiskers**
5 **extend to the maximum and minimum projections.**





**Figure 7. The annual numbers of risk days by month for heavy riming and heavy snow loading during 1980–2009 (black), 2010–2039 (red), 2040–2069 (blue), and 2070–2099 (green) under the RCP4.5 and RCP8.5 scenarios as averaged over southern (S; top row), central (C; middle row) and northern (N; bottom row) Finland.**

