# Peer review of "Heavy snow loads in Finnish forests respond regionally asymmetrically to projected climate change"

_Natural Hazards and Earth System Sciences, 2016_

## Referee Comment (RC1) · Anonymous Referee #1 · 15 Jul 2016

"General comments"

Dear authors, congratulation to this - in general - scientifically sound and well-written paper. It is about an interesting topic in climate change effects on snow loads in Finnish forests, nicely fits to the scope of the journal, represents state-of-the-art research and is in correct English. I recommend publication after some improvements. The paper could mainly benefit from (i) adding a more detailed description of the methods, and (ii) providing more quantitative measures of uncertainty, particularly for the humidity/rime load calculations.

"Specific comments"

[Figure]

- The paper could be improved in its methodological part by adding a paragraph on the two methods, G08 and FMI, respectively. In general, both methods should be explained in a detail that better supports the understanding of the results. - It would be good to explain why You actually do present the G08 method, because its results mostly correlate with dry snow loads which have little importance with respect to forest damage. If it turns out that this is of minor importance for the paper, You can consider to completely skip the G08 methods and all its results, and only mention it in the introduction. The interested reader won't probably miss it. - You should explain the FMI method with sufficient detail, and provide a meaningful measure of uncertainty for the effect of a changing (modelled and corrected/downscaled) humidity on the riming process which You state is the most important factor leading to heavy crown snow loads. The difference of humidity relative to ice and/or water appeared to be the reason for a 20 % difference in calculated maximum rime load. You should make an attempt to separate this effect from the one originating in a changing climate. - You should also justify in more detail the temporal scale transition from 3-hourly to daily (at least by providing an example). - The role of the thresholds defined by Lehtonen et al. (2014) to determine the number of the two types of risk days is not entirely clear: on the one hand, You state that they may not be suited for the whole country, on the other hand You provide their values with two decimal places (probably table 2 should be modified). I would be good to explain how these thresholds were determined, and how/why they can be applied in Your study (risk days of heavy riming vs FMI-modelled heavy riming). - You should give a short explanation in the introduction how trees are damaged by snow loads (the process(es), and how they are related to the relevant snowfall events; are there observations?), and You should give some more information about the tree species related damage risk in a changing climate.

"Technical corrections"

- Page 2, line 20: dot (".") missing between "Finland" and "In both studies" - Page 15, capture of fig. 2: s, c and n should also be denoted for southern, central and northern

(like in the capture of fig. 6). - Page 18, fig. 5: scale bar of panel section (f) is too small. Values in the map seem to not correspond with the (colors and values of the) scale bars.

Good luck!

---

## Referee Comment (RC2) · Anonymous Referee #2 · 21 Aug 2016

General comments

I liked the paper a lot. It was concise and well written. Topic is highly interesting, and it has also practical importance when thinking about forestry and how prepare the forestry sector to probable changes in the winter climate and extreme weather events.

The method used is valid and enough attention has been paid to use a set of climate scenarios; especially the bias correction has been done with plenty of thought and effort.

In Introduction there was material that belongs to Material and methods. On the other hand, aims of the work have not been given clearly in Introduction.

[Figure]

This study is built on earlier FMI work. Most of the earlier works referred are Finnish. It would be interesting to hear about earlier work done on this subject also in other parts of the world, if any. Also the importance of the topic could be broadened by telling about the snow damages internationally. Terms "forest damage" and "snow damage" could be defined, what sort of damages are we talking about.

There are also more detailed snow accumulation / unloading models available. Could you add a short explanation of these models in the paper?

Specific comments

Page 2 You talk about effects of temperature on the processes. How does the moisture affect? Effects of forest management options: tell more about these. There is something wrong with the logic of the sentence on lines 14-16. Effect of soil frost: did you discuss about this in this paper also? Line 24: would be -> will be? Line 28: please check the word order in a sentence beginning "We use..." Lines 30-31: please tell a bit more about the RCPs.

Line 9: Historical simulations? Explain better. Line 30: Delta-change method does not tell anything if reader is not familiar with this term. Explain more. Lines 10 and 11: so here you could tell a bit more about RCPs.

Page 4: There is good discussion about humidity effects here, but perhaps it should be in Discussion? Line 20: your winter period is from November to March. How well can you compare the results to earlier work, when normally the winter period is from December to February?

Page 5 Line 7: other transformations? Please list. Line 13: what do you mean "on average rather well"? Line 22: reference to this information of riming efficiency?

Do you feel that possibility to use e.g. 3h data would improve the results at all? Perhaps in case of wet snow and unloading the processes may be rather fast. And did I

understand correctly that daily precipitation was divided evenly during the day? This may also have some effect, because for sure the precipitation intensity (and other conditions during snow fall) affect. Check that you have covered this type of discussions in the paper.

Page 6 Do you have any observational data to compare the simulation results with? I understand this work is mostly about scenarios, but reliability of the modelling method should be discussed somehow and relate this to some data. If this kind of discussion is found e.g. in the earlier papers referenced, please make this clear. Line 26: Sentence beginning "However, the. . ." is complicated and should be rephrased.

Page 7 Paragraph on lines 3-6 belongs to Discussion, together with other uncertainty –discussion? Line 23: Please check the word order of the sentence beginning "Most uncertain. . .".

How does the tree species and other forest characteristics affect? You mention that they affect, but discussion could be expanded.

This would also be a good place to discuss about the model you use, and other snow accumulation / unloading models (some sort of comparison in terms of processes and time resolution, for example).

Page 8 This page has discussion that is rather long. There is some repetition when comparing to Introduction and parts of the text are rather speculative. So please consider shortening here. Especially part beginning from line 25 is a bit disconnected.

Page 11 Reference list seems adequate. I did not go through it in detail, so please make sure once more that it is in accordance with the journal instructions.

Page 14 Values in Table 2 are given with too much precision.

Technical corrections

Page 2 Line 20: a dot is missing

---

## Author Comment (AC1) · 19 Sep 2016

Journal: NHESS
Title: Heavy snow loads in Finnish forests respond regionally asymmetrically to projected climate change
Authors: I. Lehtonen, M. Kämäräinen, H. Gregow, A. Venäläinen and H. Peltola
MS No.:nhess-2016-184
MS Type: Research Article
Iteration: First review
Referee #1

*We are grateful to the Referee for the positive feedback and good suggestions which will certainly improve the manuscript. Our replies to the comments are given in "Italics" after the comments given in the beginning of this document.*

"General comments"
Dear authors, congratulation to this - in general - scientifically sound and well-written paper. It is about an interesting topic in climate change effects on snow loads in Finnish forests, nicely fits to the scope of the journal, represents state-of-the-art research and is in correct English. I recommend publication after some improvements. The paper could mainly benefit from (i) adding a more detailed description of the methods, and (ii) providing more quantitative measures of uncertainty, particularly for the humidity/rime load calculations.

*Thank you for these encouraging views. Description of the methods is in the current manuscript version rather short as we refer to an earlier study where the snow-load calculation methods have been described in detail. Nevertheless, this part could be expanded in order to provide for the readers a more deepen idea of the used methods without a need to take a look on the previous paper. Measuring quantitatively the uncertainty related particularly to the humidity/rime load calculations is not a straightforward issue. Basically, we have estimated quantitatively the uncertainty related to our projections by comparing the results based on different climate models. This range of model-based uncertainty is presented in Figs 2 and 6. It is visible that the uncertainty related to the humidity and rime load projections itself is not larger than the uncertainty related to projections in other weather variables or snow load components. However, we admit that the applied bias-correction method is not necessarily as applicable for relative humidity in freezing temperatures as for other variables and this potentially induces to the rime load projections uncertainty that is hard to measure.*

"Specific comments"
- The paper could be improved in its methodological part by adding a paragraph on the two methods, G08 and FMI, respectively. In general, both methods should be explained in a detail that better supports the understanding of the results.

*We agree that a more detailed explanation of the methods would be beneficial as then readers would not need necessarily to dig the information from the cited previous papers.*

-  It  would be good to explain why You actually do present the G08 method, because its results mostly correlate with dry snow loads which have little importance with respect to forest damage. If it turns out that this is of minor importance for the paper, You can consider to completely skip the G08 methods and all its results, and only mention it in the introduction. The interested reader won't probably miss it.

*The main reason for presenting the results for G08 method is that in the previous work of Kilpeläinen et al. (2010) this method was used and they got the result that snow loads and snow-induced forest damage in Finland are likely to decrease in the future due to global warming. As our main conclusion is quite different, we wanted to demonstrate that by using the same method, we would have got actually almost identical results, so the difference between our results is most likely due to the methodological differences, not due to the different climate change scenarios, for instance.*

- You should explain the FMI method with sufficient detail, and provide a meaningful measure of uncertainty for the effect of a changing (modelled and corrected/downscaled) humidity on the riming process which You state is the most important factor leading to heavy crown snow loads. The difference of humidity relative to ice and/or water appeared to be the reason for a 20 % difference in calculated maximum rime load.  You should make an attempt to separate this effect from the one originating in a changing climate.

*The problem here is that although after correcting the bias both in temperature and relative humidity simulations, both variables have a realistic distribution (for relative humidity in subzero temperatures this holds exactly only when considering the relative humidity with respect to ice), but the humid and dry days in the model are not necessarily distributed similarly as observed regarding the temperature distribution. There are no easy solution for this. It appears that after bias correction climate model results tend to overestimate the situations with combined temperature and relative humidity conditions favouring the rime formation. It is moreover noteworthy that in many climate models relative humidity values in cold temperatures are more or less unreliable having physically unjustified values with relative humidity exceeding 100% that raises worries considering their usefulness. To conclude, we are more suspicious of our results related to the rime loads than other snow-load components. However, we assume that in climate model results the same deficiencies are present both in the calibration period (1981-2010) and scenario periods and thus the projected changes are originating from the climate change signal. We moreover note that our projections for heavy rime loads and heavy wet snow loads indicate increases and decreases roughly over the same areas reinforcing the idea where the risk for snow-induced forest damage is likely to increase.*

- You should also justify in more detail the temporal scale transition from 3-hourly to daily (at least by providing an example).

*There are an example in the cited literature dealing this issue which could be shortly discussed here as well.*

- The role of the thresholds defined by Lehtonen et al.  (2014) to determine the number of the two types of risk days is not entirely clear: on the one hand, You state that they may not be suited for the whole

country, on the other hand You provide their values with two decimal places (probably table 2 should be modified). I would be good to explain how these thresholds were determined, and how/why they can be applied in Your study (risk days of heavy riming vs FMI-modelled heavy riming).

*This could be discussed in more detail.*

- You should give a short explanation in the introduction how trees are damaged by snow loads (the process(es), and how they are related to the relevant snowfall events; are there observations?), and You should give some more information about the tree species related damage risk in a changing climate.

*These issues could be discussed in more detail in the introduction.*

"Technical corrections"
- Page 2, line 20: dot (".") missing between "Finland" and "In both studies"

*Thank you for noting this.*

- Page 15,
capture of fig. 2: s, c and n should also be denoted for southern, central and northern (like in the capture of fig. 6).

*We agree with this.*

- Page 18, fig. 5: scale bar of panel section (f) is too small. Values in the map seem to not correspond with the (colors and values of the) scale bars.

*Thank you for noting this.*

Good luck

*Thank you*

---

## Author Comment (AC2) · 19 Sep 2016

Journal: NHESS
Title: Heavy snow loads in Finnish forests respond regionally asymmetrically to projected climate change
Authors: I. Lehtonen, M. Kämäräinen, H. Gregow, A. Venäläinen and H. Peltola
MS No.:nhess-2016-184
MS Type: Research Article
Iteration: First review
Referee #2

*We are pleased with the positive feedback and constructive comments of this Referee. Our replies to the comments are given in "Italics" after the comments given in the beginning of this document.*

General comments
I liked the paper a lot. It was concise and well written. Topic is highly interesting, and it has also practical importance when thinking about forestry and how prepare the forestry sector to probable changes in the winter climate and extreme weather events. The method used is valid and enough attention has been paid to use a set of climate scenarios; especially the bias correction has been done with plenty of thought and effort. In Introduction there was material that belongs to Material and methods. On the other hand, aims of the work have not been given clearly in Introduction. This study is built on earlier FMI work. Most of the earlier works referred are Finnish. It would be interesting to hear about earlier work done on this subject also in other parts of the world, if any. Also the importance of the topic could be broadened by telling about the snow damages internationally. Terms "forest damage" and "snow damage" could be defined, what sort of damages are we talking about. There are also more detailed snow accumulation / unloading models available. Could you add a short explanation of these models in the paper?

*We are thankful for these many suggestions which could be used to improve the paper, particularly the Introduction section.*

Specific comments

Page 2 You talk about effects of temperature on the processes. How does the moisture affect? Effects of forest management options: tell more about these. There is something wrong with the logic of the sentence on lines 14-16. Effect of soil frost: did you discuss about this in this paper also?

*Thank you for these comments. Soil frost was not discussed nor studied in this paper.*

Line 24: would be -> will be?

*Ok*

Line 28: please check the word order in a sentence beginning "We use:::"

*We will check it.*

Lines 30-31:  please tell a bit more about the RCPs.

*Ok*

Line 9: Historical simulations? Explain better.

*Historical simulations refer to simulations over historical time period with historical forcing data (emissions/concentrations/land-use change).*

Line 30: Delta-change method does not tell anything if reader is not familiar with this term. Explain more.

*Thank you for this comment. Surely an explanation should be included. Basically, in the delta-change method the distribution is only shifted so that the mean corresponds to the observed value and optionally also the variability can be corrected similarly.*

Lines 10 and 11: so here you could tell a bit more about RCPs.

*Yes, and I think this is a better place for that than the introduction section.*

Page 4:  There is good discussion about humidity effects here, but perhaps it should be in Discussion?

*The discussion here is used to justify the selected approach in correcting the simulated relative humidity values.*

Line 20:  your winter period is from November to March.  How well can you compare the results to earlier work, when normally the winter period is from December to February?

*We chose to show the projected changes in weather variables over this extended winter period because the snow damages are not restricted to the December-February period. In Finland, the projected warming is most pronounced in midwinter, but in general November and March are projected to warm only slightly less than the traditional winter months (i.e. December, January and February). Anyway, you are right, the projected changes for the December-February period would be mostly just slightly strengthened compared to the November-March period.*

Page 5 Line 7:  other transformations?  Please list.

*The possible snow load type transformations are:*
   - *Change of dry snow into wet snow (due to rain or melting)*
   - *Change of frozen snow into wet snow (due to rain or melting)*

- *Change of wet snow into frozen snow (due to freezing)*

Line 13: what do you mean "on average rather well"?

*The correlation coefficient was 0.82 for the total snow load over four 30-year periods.*

Line 22: reference to this information of riming efficiency? Do you feel that possibility to use e.g. 3h data would improve the results at all? Perhaps in case of wet snow and unloading the processes may be rather fast. And did I understand correctly that daily precipitation was divided evenly during the day? This may also have some effect, because for sure the precipitation intensity (and other conditions during snow fall) affect. Check that you have covered this type of discussions in the paper.

*Yes, you have understood correctly. We feel that the possibility to use e.g. 3 hourly data would improve the results a lot but mainly in individual cases. Based on our earlier work, we feel that the broad-scale climatological characteristics over a long time period are fairly similar nevertheless daily or hourly data is used.*

Page 6 Do you have any observational data to compare the simulation results with? I understand this work is mostly about scenarios, but reliability of the modelling method should be discussed somehow and relate this to some data. If this kind of discussion is found e.g. in the earlier papers referenced, please make this clear.

*We have this kind of discussion in our earlier paper, although the amount of observational data is limited.*

Line 26: Sentence beginning "However, the:::" is complicated and should be rephrased.

*Thank you for this comment.*

Page 7 Paragraph on lines 3-6 belongs to Discussion, together with other uncertainty–discussion?

*We disagree with this because inspecting the model-based uncertainty in the projected change is one of the main objectives of this study and the results of that inspection thus clearly belong into a Results section. Unlike other uncertainty sources related to this study, this is the only one that we study quantitatively. However, as you noted in your general comments, the aims of this work should be presented more clearly in the Introduction section.*

Line 23: Please check the word order of the sentence beginning "Most uncertain:::".

*We will check it.*

How does the tree species and other forest characteristics affect? You mention that they affect, but discussion could be expanded.

*This discussion can be expanded.*

This would also be a good place to discuss about the model you use, and other snow accumulation / unloading models (some sort of comparison in terms of processes and time resolution, for example).

*This is a good idea.*

Page 8 This page has discussion that is rather long.  There is some repetition when comparing to Introduction and parts of the text are rather speculative.  So please consider shortening here. Especially part beginning from line 25 is a bit disconnected.

*We agree that this part of the discussion can be compressed and shortened.*

Page 11 Reference list seems adequate.  I did not go through it in detail, so please make sure once more that it is in accordance with the journal instructions.

*We will check it once more.*

Page 14 Values in Table 2 are given with too much precision.

*We can leave out the second decimal.*

Technical corrections

Page 2 Line 20: a dot is missing

*Thank you for noting this. Well done from the referees as both of them noticed this same error.*